# Does Cultural Difference Matter on Social Media? An Examination of the Ethical Culture and Information Privacy Concerns

**Sangmi Chai**

Ewha School of Business, Ewha Womans University, 52 Ewhayeodae-gil, Seodaemun-gu, Seoul 03760, Korea; smchai@ewha.ac.kr; Tel.: +82 2 3277 2780

**Abstract:** While social media has become a very popular tool for sharing information and news worldwide, the ethical culture of the users emerged as a significant issue in cyber space. This research investigates the role of perceived ethical culture and information privacy concerns on social media behaviors. More importantly, this study investigates the role of cultural difference in the relationship among those factors. Based on the study results of U.S. and Korean social media users, this study found ethical culture to be positively associated and information privacy concerns negatively associated with users' information-sharing behavior on social media. In addition, the study results indicated that the size of the impact of the two facts are varied between the two countries. This study's results direct that users' perceived ethical culture and privacy concerns are important factors affecting social media users' information sharing. However, these factors could have a different impact with cultural differences.

**Keywords:** ethical culture; information privacy concerns; social media; information-sharing behavior

## 1. Introduction

These days, due to the usage of emerging information technologies, people are affected significantly by online interactions. Various forms of Internet-based communications, online communities, social media, and online discussion boards show a distinct subculture influencing people's behaviors, both online and offline. Because of people's heavy Internet usage, social media has a significant impact on their decision-making for product purchasing, participation in political movements, and participation in online communities. Especially in the e-commerce environment, people search and share product information in social media by posting reviews, influencing consumers' product purchasing behaviors [1–3]. People search product reviews in social media to reduce the risks related with product purchasing, such as refunds, complaints, and exchanges [4]. Ethical culture plays a significant role in stimulating social media users' activities in sharing information as well as knowledge [5]. Based on above discussion, this study considers culture as one of the most important preconditions for sound information sharing on social media. In this study, culture in social media is examined as two perspectives: as one of influential factors, stimulating information sharing, and as a factor having moderator effects on each influential factor.

Through social media, people go beyond the existing maintenance of social networking to form new relationships while sharing their concerns, political perspectives, and hobbies with strangers [6]. Since social media is an Internet service based on human networks and is a virtual society where many people are gathered, cultures can be formed. In addition, such cultures on social media can be affected by offline situations because the range of offline human relations are similar to those of online human relations, resulting in offline cultures being able to affect online cultures, too. According to

previous studies, a major motive for using social media is to maintain and expand human networks [7,8], and many people use online networks to continuously communicate with offline acquaintances in many cases [9]. Therefore, this research investigates if cross-national differences involving offline sociocultural characteristics should affect social media environments as well.

Most of contents of social media come from individual participants. Thus, the cases of personal information misuses and invasions are rapidly increasing and the damages due to online unethical behaviors, such as cyber bulling and malicious comments, are easily found. According to existing research, privacy concerns are identified as one of the main factors that hinder online community activities [10,11].

According to previous studies, the organization's ethical culture is discussed as an important factor in promoting information sharing among members [5,12]. However, few papers have studied ethical culture directly in cyber environments. This study intends to empirically study the information-sharing behavior of users on social media in two main aspects. The first objective is exploring the role of ethical culture and the impact of information privacy concerns on social media users' information-sharing behavior. In addition, the research aims to explore the effects of users' offline socio-cultural characteristics on information-sharing behavior. To archive the objective, social media users in two countries (U.S. and South Korea) are explored to find out the influence of cultural difference on ethical culture, information privacy concerns, and information-sharing behaviors.

## 2. Ethical Culture

The primary purpose of my study is to cite the concept of ethical culture to see how it affects information sharing on the social media according to cross-national differences. Ethical culture is defined as "a subset of organizational culture, representing a multidimensional interplay of various formal and informal systems of behavior control that are capable of promoting ethical or unethical behavior" (p.12) [13]. In a study conducted by [5], ethical culture was redefined as norms that users share in the social media environment based on the definition of [13]. My study adopts the concept of ethical culture that there are norms and values shared in a social media community. Thus, the existence of ethical codes of conduct was identified, and the fact that these ethical codes of conduct can have positive effects on knowledge-contribution behaviors [5]. Since online ethics can be delivered in abstract meanings, what are acknowledged as moral and ethical behaviors in social media environments should be checked. In general, ethical issues in social media environments can be examined in terms of identity, surveillance, motives for use, user exploitation, and privacy [3,5,12,14–18]. Social media is a space where personal information and areas of interest are disclosed, and refined knowledge is shared. However, social media as such can be utilized for commercial or impure purposes, instead of purposes expected by people, to have unethical effects due to untrue identities and misused information. For instance, a study identified how social media users' identities are constricted and co-constructed from an ethical point of view [19]. It emphasized that social media identities may affect offline identities too and importantly deal with wrong role playing with identities that are not real identities due to relationships on social media because of the reflexive nature of identities. Additionally, while reporting "shilling", which is acts of posting fake comments, blogs, and social media posts after being paid by enterprises, the New York Times dealt with ethical issues in terms of motivations for its use [12]. These ethical issues relating to identities and motivations for use can undermine the credibility of information on social media. The utilization of social media as a surveillance tool of employers also appears as an ethical problem. Enterprises monitor social media to understand the privacy and propensities of employees, and utilize the information obtained this way to influence individuals' job performance [17]. In such cases, social media users may feel that they are watched. In addition, prior studies found ethical issues in social media. These include a case where personal information on the social media is used by ranking friends' applications and a case where personal information is used without permission by systems that automatically recommend users to form friends through user exploitation [20,21].

Proper use of social media can provide abundant social, business, and personal benefits. However, the use of social media should be based on the belief that the information and knowledge

on social media are used ethically. Previous studies have suggested that the role of ethics is important to gain people's trust, and ethics is essential [22,23]. Based on the above discussion, this research addresses ethical issues on social media through socio-technical approaches.

## 3. Information Privacy Concerns

Westin defines information privacy as "the claim of individuals, groups or institutions to determine of themselves when, how, and to what extent information about them is communicated to others (p.7)" [24]. I have mentioned the ethical issues on the diverse aspects of social media earlier. According to past research, privacy concerns of individuals can vary depending on how they are regulated by the state [17], while the privacy policies and data protection laws for websites differ by country. For instance, in the case of the U.S., the federal government's regulations on data privacy are not stronger when compared to other countries. Since the U.S. place great emphasis on the use of data and has different laws and regulations by industry, the rules and regulations are applied complexly, depending on the U.S. website. Although the Federal Trade Commission (FTC) stipulates "business privacy laws" and does not strongly demand privacy policies to businesses, it prohibits deceptive practices. In addition, the Health Insurance Portability and Accountability Act of 1996 (HIPAA), which deals with health-related information, and the Children's Online Privacy Protection Rule (COPPA), which deals with online child information, are strictly applied. Unlike the U.S., most EU countries strongly prohibits the collection of personal data without obtaining the individuals' express consent pursuant to General Data Protection Regulation and requires that data should be directly requested to the entity when the data are utilized. When collecting and utilizing others' data, information on the purpose should be provided and the data cannot be utilized for other purposes than the relevant purpose. South Korean privacy regulations also require prior consent to data collection pursuant to "the Act on the Promotion of Information and Communications Network Utilization and Information Protection, etc.", and provide clear criteria for the validity of the consent form. Protection policies for privacy by country can also play an important role in forming users' perceived information privacy concerns. However, there are many differences in personal information protection in terms of legal aspects of Korea and the United States. Since Korea enforces personal information protection with three major laws, the Personal Information Protection Act, the Network Communication Act, and the Credit Information Protection Act, the nationwide aspect is stronger than that of each industry. In contrast, in the case of the U.S., each state has a different personal information protection law, and it is a legal system that is applied differently by industry (like the FTC's business privacy law and HIPPA) and by target (COPPA). In addition, for the protection of personal information, the focus is on compensation for actual victims in the event of personal information leakage. However, in the case of Korea, the elements stipulated by the Personal Information Security Act are stipulated in great detail up to the guideline level, and it is very important not to violate these laws from the standpoint of companies, and it is difficult to punish any matters other than those stipulated in the law. Accordingly, the Personal Information Protection Act in Korea makes compensation for leakage of personal information insignificant or very difficult if it is not a violation of the Personal Information Protection Act.

Beyond the aspect of the classification of the national characteristics that considered centering the cultural characteristics, whether the effects of information privacy concerns vary with cross-national differences, considering all external environments such as society, laws, and administration, have been examined. A recent 2019 survey conducted by the Pew Research Center indicates that 81% of respondents are concerned about their online information privacy and privacy risk. Another study also found out that 26% of respondents disclosed false post and false personal information, such as false names, education, and regional information to protect privacy along with the fact that many of teenage users in the U.S. disclose their actual personal information [25]. Based on this report, information privacy concerns can change information-sharing behaviors. Therefore, it should be examined whether lowering the level of privacy concern can act as a practically important way for the creation of a healthy social media environment for active information sharing.

## 4. Research Model and Hypothesis

The formation of an ethical culture on social media is important for sharing information continuously among social media users. People want to use shared information in proper ways. However, abuse of shared information happens a lot on social media, like unethically modified information or false information used to propagate people, eventually leading to damage to the users. Some studies indicated that interrelationships between people in online environments can promote information-sharing behaviors when the interrelationships are based on trust [26,27]. Shared value is important to enhance trust in online environments [28]. Shared value means people's common beliefs about other people' behaviors or goals and researchers explain that ethics play a very important role for shared value [29]. Eventually, people share their information on social media based on mutual trust and mutual trust requires the belief that those that share information will not cause damage to them as information is properly shared [30,31]. A study argued that a variety of ethical behaviors (distributive justice, procedural justice, and cooperativeness) within organizations has positive effects on tacit knowledge sharing through trust among people [31]. In addition, the research indicated that trust relationships are established, and the level of information sharing is enhanced in the relationship between the supplier and the buying firm when unethical behavior has been reduced [30].

In the present study, I examine whether or not ethical behaviors in cyberspace also affect information-sharing behaviors like ethical behaviors in physical society. In social media environments where personal information is extensively disclosed, although information can be shared easily and without any restriction, many ethical dilemmas exist for social media users. If they do not have belief that all of the users do use shared information in an ethical way, they may be afraid to share information. However, common rules in promises for behaviors in groups certainly exist and the ethical culture implicitly perceived on social media is expected to have a positive effect on the information-sharing behaviors.

Based on these discussions, my research posits the hypothesis that users' perception of ethical culture can positively affect information-sharing behaviors in social media environments, as with the relevant results in offline environments. Based on the assumption that ethical culture increases a user's information-sharing behaviors, this study posits the following hypothesis.

**Hypothesis 1 (H1).** *Perceived ethical culture is positively associated with information-sharing behaviors of social media users.*

Social media is faced with many problems relating to users' information privacy because users share personal information a lot on social media. Existing previous studies indicated that privacy concerns are closely related to exposure of personal information on social media [16,18]. They stated that social media users with higher privacy concerns showed the lower degrees of exposure of personal information [18]. The research of [16] suggested that social media users who feel relatedly low privacy concerns have a tendency to share information for the benefits that can be obtained by exposing personal information in many cases.

A prior study presented an idea that people may not consider social media as public space, so that when people post personal information on social media [32], it may bring privacy invasion into the online space. In addition to personal information disclosure on social media, overall information sharing is expected to be affected by privacy concerns as well as personal information exposure because it is directly exposed and shared through the domain of individual accounts.

Based on these discussions, this study suggests the following hypothesis. The exposure of personal information on social media as well as overall information sharing on social media are expected to be affected by privacy concerns because information is also directly exposed and shared through the domains of personal accounts. Therefore, this study presents the following hypothesis.

**Hypothesis 2 (H2).** *Information privacy concerns are negatively associated with information-sharing behaviors of social media users.*

A study has described a popular culture as a way of using a product in specific ways [33]. Although diverse people around the world use social media, they do not share the same thoughts and beliefs. Users placed in these diverse conditions and environments may perceive different ethical cultures [34]. Previous studies revealed differences in information-sharing behaviors in online review systems due to cross national differences and argued that information-sharing behaviors were closely associated with cultural elements [35,36]. Their study argued that writing online movie reviews was affected by cultural differences, such as differences in the social norms and attitudes among countries, and indicated that significant results were identified through investigations with the U.S., China, and Singapore. The study results revealed that the reviews of the U.S. are more extreme than those of China and Singapore [36]. A study indicated that different cultural elements (e.g., collectivism, competitiveness, attention paid to power and hierarchy, culture-specific preferences, etc.) led to significant differences in knowledge-sharing behaviors in enterprises [22]. In addition, the degree of privacy concerns of individuals can vary with personal information protection systems and regulations, which are different by country [14].

Therefore, it is very important to examine whether differences between users' cultural difference (the U.S. and South Korean users) affected their perceived ethical culture and information privacy concerns, as well as their information-sharing behaviors, or not. Based on these discussions, I posit the following hypotheses.

**Hypothesis 3 (H3).** *The size of impact of ethical culture on information-sharing behaviors on social media is different with users' cultural background.*

**Hypothesis 4 (H4).** *The size of the impact of information privacy concerns on information-sharing behaviors on social media is different with users' cultural background.*

## 5. Methodology and Research Results

This research applied a survey questionnaire method to collect the data, developed from previous literature. Since the survey is the most suitable research method in human behavior studies, with experiments and observation, this study adopted a structured survey method to identify social media users' perceptions regarding online privacy concerns, ethical culture, and information-sharing behaviors, as prior studies did [5,15,37]. The questionnaires are presented in Appendix A. Targeting social media users in the U.S. and Korea, the questions regarding ethical culture and online information privacy concerns were asked in a way to rate the degree of agreement. The six measurement items of ethical culture were developed from [5]. The eight measurement items for online information privacy concerns were adopted from [15]. The eight construct measurements for information-sharing behavior were developed from [37]. All construct measurements were modified in the context of social media and used with a seven-point Likert scale (1–7). A factor analysis was conducted and the factor loadings for all the measurement items are presented in Table 1. The factor analysis of all the measurement items indicated that they are all properly correlated with factors such as all six measurement indicators of ethical culture, which was higher than 0.6 with ethical culture but of low correlation with other factors.

**Table 1.** All factor loadings of the measurement items from the factor analysis.

| Items | EC(Ethical Culture) | IPC (Information Privacy Concerns) | ISB (Information-Sharing Behaviors) |
|-------|---------------------|------------------------------------|-------------------------------------|
| EC1 | **0.874** | 0.052 | 0.308 |
| EC2 | **0.787** | 0.187 | 0.264 |
| EC3 | **0.820** | 0.262 | 0.077 |
| EC4 | **0.799** | 0.136 | −0.073 |
| EC5 | **0.800** | 0.212 | 0.143 |
| EC6 | **0.812** | 0.111 | 0.190 |
| IPC1 | −0.156 | **0.807** | −0.144 |
| IPC2 | 0.217 | **0.832** | −0.068 |

| | | | |
|---|---|---|---|
| IPC3 | 0.275 | **0.873** | −0.184 |
| IPC4 | 0.269 | **0.836** | −0.058 |
| IPC5 | 0.211 | **0.904** | −0.097 |
| IPC6 | −0.021 | **0.899** | −0.107 |
| IPC7 | −0.011 | **0.921** | −0.020 |
| IPC8 | 0.299 | **0.803** | −0.038 |
| ISB1 | 0.130 | 0.141 | **0.870** |
| ISB2 | 0.230 | −0.143 | **0.830** |
| ISB3 | 0.259 | −0.078 | **0.891** |
| ISB4 | 0.133 | −0.3331 | **0.814** |
| ISB5 | 0.306 | −0.045 | **0.843** |
| ISB6 | 0.292 | −0.170 | **0.810** |
| ISB7 | 0.224 | −0.165 | **0.844** |
| ISB8 | 0.243 | −0.202 | **0.877** |

The pilot study was conducted by having in-depth interviews with heavy social media users and IT professionals. By reflecting suggestions and feedbacks from the interview, I focused on revising questions in the context of social media to improve the reliability as well as validity. By collecting more than thirty responses in the U.S., I clarified questions in the survey based on feedbacks. I also conducted a pilot study in Korea and revised the survey questionnaires with feedbacks although I did not drop any measurement items in the U.S. nor Korea.

I distributed the questionnaire to social media users in the U.S. and Korea and collected a total of 406 responses. To distribute the survey, an online survey was created and the link to the survey was distributed to online communities in universities in Korea and the U.S.

After taking out incomplete surveys, there were 389 usable responses. Thus, I used 200 responses from the U.S. and 189 responses from Korea for the data analysis. Most respondents were college students who were frequently using the Internet as well as social media; males gave 198 responses (50.9%) and females 191 responses (49.1%). The study sample demographic is presented in Table 2. Because I applied only the survey methodology, I conducted a common method bias test by using Harman's single factor test. Following all steps from the research of [38,39], I checked all the eigenvalues by performing an unrotated factor analysis. I found the evidence that the sum of variances of a single factor and the first factor was not greater than 20% of the variance in all factors. Therefore, I concluded that there was no common method bias in this data.

**Table 2.** Sample demographics.

| Nationality | Frequency | Percentage |
|---|---|---|
| American | 200 | 51.41% |
| Korean | 189 | 48.59% |
| Gender | | |
| Male | 198 | 50.90% |
| Female | 191 | 49.10% |
| Age | | |
| < 20 | 4 | 1.03% |
| 20–29 | 188 | 48.33% |
| 30–39 | 126 | 32.39% |
| 40–49 | 44 | 11.31% |
| 50+ | 27 | 6.94% |
| Education | | |
| High School Graduates | 12 | 3.09% |
| Bachelor's degree | 368 | 94.60% |
| Masters' degree | 9 | 2.31% |

For the data analysis, this research used the partial least square techniques. By performing bootstrapping, PLS has a strong advantage by establishing both measurement models and full

structural models [40]. For assessing the reliability and construct validity, I calculated Cronbach's Alpha and the composite reliability (CR). In addition, I also examined the average variance extracted (AVE). All these values indicate a good construct reliability, which are presented in Table 3. I also conducted a factor analysis by PLS and presented all factor loadings in Table 4. For assessing the discriminant validity, I compared the correlations of the constructs and square roots of the AVE following the research of [41]. Since the square roots of the AVE of each variable were greater than the correlations of any other variable, the construct measurements show the acceptable discriminant validity following the Fornell–Larcker criterion [41], which are presented in Table 4. As the discriminant validity test result indicated, the factors used in this study shows a lower than 0.5 coefficient with other factors than itself. The coefficients are presented in Table 4; the numbers in bold are greater than the off-diagonal elements so that there is low chance or multicollinearity among the factors. The cross-loadings are where an indicator's outer loading on a variable in Table 2 are greater than all its cross-loadings with other variable, calculated make sure that the discriminant validity is acceptable. Finally, I also checked the heterotrait/monotrait ratio of the correlations (HTMT) and confirmed that all values are smaller than 0.9, validating the discriminant validity.

**Table 3.** Reliabilities and validity.

| Constructs | Cronbach's Alpha | CR | AVE |
|---|---|---|---|
| Ethical Culture | 0.882 | 0.840 | 0.675 |
| Information Privacy Concerns | 0.849 | 0.823 | 0.598 |
| Information-Sharing Behavior | 0.888 | 0.912 | 0.766 |

**Table 4.** Discriminant Validity.

| Variables | Ethical Culture | Information privacy concerns | Information-Sharing Behavior |
|---|---|---|---|
| Ethical Culture | **0.822** | | |
| Information privacy concerns | −0.002 | **0.773** | |
| Information-Sharing Behaviors | 0.419 | −0.163 | **0.875** |

By conducting bootstrapping procedures, I found statistical significances among three constructs. My research results supported Hypothesis 1: Ethical culture is positively associated with information-sharing behaviors of social media users. The research results present a significant relationship statistically with a path coefficient of 0.399 and a t-score of 2.66 at a 0.01 level of significance. Hypothesis 2: Information privacy concerns are negatively associated with information-sharing behaviors of social media users, was also supported by my research results. A statistically significant relationship was confirmed with a path coefficient of −0.238 and a t-score of 2.05 at the 0.05 level of significance.

Hypothesis 3: The size of the impact of ethical culture on information-sharing behaviors on social media is different with users' cultural background; and Hypothesis 4: The size of impact of information privacy concerns on information-sharing behaviors on social media is different with users' cultural background, were supported by my data analysis results. For examining the moderating effects on Hypotheses 3 and 4, this research conducted a subgroup analysis between U.S. and Korean social media users with PLS. It investigated whether the differences in path coefficients on the relationships between each construct are significant or not based on the research of [42]. Following the research of [43], it compared the statistical path between the two groups. It summed up the square root of the standard error estimates for the structural models of the two groups and computed the square root of the sum number at the bottom. It computed the difference of the path coefficients between two subgroups. By calculating the numbers at the top and bottom, it provided the t-statistics, which can determine a statistical significance in the differences between the two path coefficients of the two subgroups [43].

The path coefficients for ethical culture and information privacy concerns with information-sharing behaviors in the U.S. social media users group were 0.213 and −0.426, respectively, with t-scores of 2.18 and 3.87, respectively, which were statistically significant at the $p < 0.05$ and $p < 0.01$ level in each relationship. A total of 200 responses (51.41% of the total sample) was from the U.S. social media users. The path coefficients for ethical culture and information privacy concerns with information-sharing behaviors in the Korean social media user group were 0.418 and −0.207, with t-scores of 3.63 and 2.10, respectively, which were statistically significant at the $p < 0.01$ and $p < 0.05$ level in each relationship. A total of 189 responses (48.59% of the total sample) was from Korean social media users.

When comparing the path coefficients between the U.S. and Korean social media users, I found statistically significant differences between ethical culture and information-sharing behaviors. The t-value of the comparison paths for ethical culture and knowledge-sharing behaviors was 2.93, which was statistically significant at the 0.01 level. Therefore, this research results support Hypothesis 3. The relationship between ethical culture and information-sharing behaviors is differentiated depending upon the culture of the social media users. The t-value of the comparison paths for information privacy concerns and information-sharing behaviors was 3.11, which was statistically significant at the 0.01 level, supporting Hypothesis 4. Table 5 summarizes my research results for the moderating effects on the culture of social media users. All of my hypotheses are thus supported by the research results and Table 6 summarizes the study results.

**Table 5.** Research results of moderating effects.

| Constructs | U.S. Subgroup $R^2 = 0.204$ (200) | | Korean Subgroup $R^2= 0.169$ (189) | | Statistical Comparison of Paths |
|---|---|---|---|---|---|
| | Standardized Path Coefficient | t-Value | Standardized Path Coefficient | t-Value | t-Value |
| Ethical culture →Information-sharing behavior | 0.213 | 2.18 * | 0.418 | 3.63 ** | 2.93 ** |
| Information privacy concerns→Information-sharing behavior | −0.426 | 3.87 ** | −0.207 | 2.10 * | 3.11 ** |

\* 0.05 significance, \*\* 0.01 significance

**Table 6.** Summary of the research results.

| Hypothesis | Standardized Path coefficient | t-Value | Results |
|---|---|---|---|
| H1 | 0.399 | 2.66 ** | Supported |
| H2 | −0.238 | 2.05 * | Supported |
| H3 | U.S.: 0.213, Korea:0.418 | 2.93 ** | Supported |
| H4 | U.S.: −0.426, Korea: −0.207 | 3.11 ** | Supported |

\* 0.05 significance, \*\* 0.01 significance.

## 6. Discussion and Conclusions

As knowledge sharing among users is important in social media environments, the present study searched for factors that can affect active information sharing. My research began by studying the concept of ethical culture based on promises and norms among users on social media as a key factor, investigating whether I can provide a realistic direction for creating ethical environments on social media through the results. Through the results of the present study, the following implications were found.

First, users who share information through social media are more active in sharing information when they feel that there is a strong ethical culture in their social media communities. However, the online privacy concerns have a clear negative impact on their information-sharing activities. The

results of the study also revealed that the positive influence of the ethical culture felt by the user is a major factor in promoting the user's information sharing rather than the magnitude of the negative impact of privacy concerns on the user's information-sharing behavior.

The research results suggest that it is important for users to recognize that there is a set of ethical rules or codes shared among users, and to recognize this as a culture that exists in the user community in order to promote information sharing in social media. In addition, providing a strong belief that privacy can be protected online when sharing various information has a very positive effect on users' information-sharing activities. The study of Chai and Kim discussed the positive role of ethical culture on bloggers' knowledge-contribution behaviors [37]. This study's results empirically confirm that users' perceived ethical culture has a positive effect on not only knowledge creation but also users' information-sharing behaviors as well. In addition, the study results revealed the role of ethical culture in an online environment, social media, while prior studies examined ethical culture in an offline organizational setting or inter-organizational environment [30,31]. More importantly, this study's results found the relative size of the effect between ethical culture and privacy concerns. Most of the past research examined the effect of privacy concerns [16,18], but this study provides evidence that an ethical culture in an online community can offset the negative effect of user-perceived privacy concerns regarding information-sharing behaviors. Secondly, it was found that the influence of the two factors, which are perceived ethical culture and online privacy concerns, on the user's information sharing differs according to the user's cultural background. As a result of this study, the path coefficient of the research model appeared differently in the user groups of South Korea and the U.S. The positive impact of perceived ethical culture was very large for Korean users, while the negative impact of perceived concerns was larger than the ethical culture for users in the U.S. In other words, in Korea, the influence of an ethical culture in the user's community is more important, and in the U.S., the privacy concerns felt by individual users compared to the ethical culture felt by the community have a stronger influence on the user's information-sharing behavior. Since there are few studies that examine cultural differences in online information sharing, as most of studies discuss the role of cultural difference in offline environments [44], the findings of this study can fill the gap in cultural difference between offline information sharing and online information sharing. In addition, prior studies discussed the difference only in information sharing [22,35,36], so that there are limitations to explain the reason why the difference exist in information sharing. This research examined the moderating effect of the explanatory variables, which are ethical culture and privacy concerns on information-sharing behaviors. More importantly, the findings of this study indicated that the two factors have different levels of impacts on information sharing according to the users' cultural background. These findings can contribute to reducing the limitation of prior literature.

The research results show that the culture to which the user belongs should be considered as an important factor in studying various models of users' behavior on social media. The behavioral model suggested by many studies is often studied in a specific country but proposes a universal model that attempts to explain most of the online users. However, if we consider the cultural background to which the user belongs, the existing research results, as suggested by prior studies, may vary depending on the cultural background of the users.

This study also has major research implications in the development of various computer systems equipped with Artificial Intelligence (AI) with predictive models that have been spotlighted recently. AI algorithm-based computer systems are developed based on existing data and applied to various models that can be used in areas requiring human judgments. However, my research shows that the factors that influence human behavior or judgment are very diverse and can also be varied depending on the culture in which they belong so that it is very hard to construct a universal model that can be applied to every society and every user. As the study of Kalimeri and Tjostheim suggested, people show different concerns about AI based on their belonged group of society, even in the same country [45]. Therefore, this study gives an implication that a system that can be used for human behavior or judgment should consider the social and human behavior patterns properly so that various kind of research studies, including behavioral studies, should be carried out; moreover, they need to be designed to consider a socio-technical approach.

This study has some limitation related to the research sample as only two countries' users were compared to assess cultural differences. To generalize the research model, a future study investigating a greater number of countries needs to be carried out. In addition, a study applying data analytic methods needs to be conducted, since this study's results depended on users' perception due to the limitation of the survey research, a future study with hard data analytics, such as online text analysis and network analysis, would be necessary to find out the hidden factors.

**Funding:** This research received no external funding.

**Conflicts of Interest**: The author declares no conflict of interest.

## Appendix A

**Table A1.** Measurement items.

| Factor | Measurement Items |
|---|---|
| Ethical Culture (EC) | The important concern for social media users is the good of all the people in the social media community. |
| | Social media users look out for each other's good. |
| | Social media users are expected to follow their own personal and moral beliefs. |
| | It is very important to use social media ethically. |
| | It is expected that social media users will always do what is right for the other social media users and public |
| | The average users use social media ethically. |
| Information Privacy Concerns (IPC) | It usually bothers me when social media companies ask me for personal information when I use social media. |
| | I am concerned about threats to my personal privacy when I use the social media. |
| | It is important for me to protect my privacy on my social media. |
| | When other social media users ask me for personal information, I think twice before providing it. |
| | When I use social media, personal privacy is important. |
| | It usually bothers me when other social media users ask me for personal information. |
| | It bothers me to give personal information to so many people. |
| | I am concerned that social media companies are collecting a lot of personal information about me. |
| Information-Sharing Behavior (ISB) | I frequently visit other social media to get information. |
| | I frequently leave my feedback/comments on other social media. |
| | I spend some time on my social media to update new information. |
| | I update my social media regularly. |
| | I frequently share my experience and information with other social media users. |
| | I provide my useful information at the request of other social media users. |
| | I share my information from my education or training with other social media users. |
| | I post useful documents or files on my social media to share with other social media users. |

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
