# Peer review of "Does Cultural Difference Matter on Social Media? An Examination of the Ethical Culture and Information Privacy Concerns"

_sustainability, doi:10.3390/su12198286_

Round 1

Reviewer 1 Report

The article is very interesting. In order to increase the quality of the paper I suggest to do the following changes:

  • present the Fornell–Larcker criterion (it is required in PLS method),
  • add discussion with the results of other researchers,
  • describe limitations and future direction,
  • present the questionnaire used in this research.

Reviewer 2 Report

This paper focuses on the cultural differences between the US and Korean citizens who use social media networks. The researcher investigates perceptions of ethical culture as defined by Trevino et al. and privacy by posing four hypotheses. Authors perform a pilot student, generate hypotheses, and answer the questions using a combination of survey questionnaires, Harman's Single Factor Test, unrotated factor analysis, and the partial least square techniques. From this investigation, it has been claimed that there is no bias in the dataset. It has been found that their culture shapes users' behavior on social media networks. The paper further warns that ethical cultures may act as a confounder when building models and drawing conclusions from intercultural data.

Strengths:

  • The purpose of the research was clearly identified and supported.
  • The aims and hypotheses have been identified and clearly stated. Sections 1, 2, and 3 set a solid foundation and theoretical framework to support the hypotheses. The hypotheses have been accepted and/or rejected with statistical verification.
  • The target cultural populations were clearly identified. The sample size was collected over social media and removed incomplete surveys.

Weaknesses:

  • The discussion does not reposition itself back in literature or related works. No future research was mentioned.
  • Submission needs to be check for grammar and other minor errors. A few sentences and ideas seem too convoluted, please separate appropriately.
  •  The bounds (age, frequency of use, etc.) of the survey could be identified.

Minor issues:

  • Abstract. Rephrase the sentence starting with "By comparing..."
  • Correct spacing and tabs in tables.
  • For line 34, "Besides commerce, social media widely used..." please provide a citation to verify this claim.
  • Provide citation/reference for the definition of culture.
  • Line 50 -- Ranges -- Range
  • Line 59, what privacy concerns are pointed out?
  • The titles for EC, IPC, and KSB can be named or referenced. The citation is there for reference; however, it would be easier to understand if those measurement titles were included.
  • Line 77 -- medal -- media
  • Line 82-83 -- provide a citation for this claim.
  • Line 91 -- put "shilling" in quotes
  • Line 96 -- grammar check
  • Line 99-100 -- grammar check
  • Section 3, please discuss the Korean and US regulations compare. This can be inferred, but these differences are key to the experiments and understanding of the results. The paper cannot be accepted without this comparison.
  • Line 154 -- Grammar
  • Line 218 and Table 1: Provide more details on the range of the Likert scale. What is the range for the factor analysis? -1 to 1? Describe the significance of Table 1 in more detail. Specify factors meaning before Table 3.
  • Consider bias for other categories than gender if you have collected that information as well.
  • Table 3: If no values are present for that item, consider placing an indicator. Move the note outside of the table bounds.
  • Make the findings of the Hypotheses stand out. They seem clustered and lost within the paragraphs.
  • Create a table to reference the data from the findings of the hypotheses.
  • Provide a sample of questions asked or add to the appendix.
  • For references, watch the date formats and italics in the titles.
  • Please watch the format of citations for consistency. 
  •  

Reviewer 3 Report

Dear author/s,

I consider your paper as up-to-date and quality processed. Mainly the theoretical parts are written well, however, I have several remarks related to research design and justification and explanation of some parts of the text.

l. 208-210 Stylistics imperfection in the hypothesis formulation (H3, H4) - avoid using "social media will be different". This can be corrected with more precise and suitable words.

l. 212 You should justify why you used a questionnaire, what are the benefits of such an approach (for example the content analysis of the communication published on social media may bring more credible results?)

l. 214-217 When defining factors of EC, resp. IPC and KSB you should be more specific (explanation of the factors, not only literature reference) because we do not know what exactly was used in your questionnaire. Moreover, it would be useful to attach the questionnaire in the attachment of your paper.

l. 220 Table 1 - KSB was not defined. Besides this, in the source referred in the paper the term KCB (contribution behavior) is used. Why do you use something else?

l. 228-237 In relation to the description of the research paper - information on the date and main specifics about the delivery of the questionnaire to the respondents is missing. Also, the sampling strategy and its justification should be described.

l. 292 and next - Discussion with other studies is missing.

Thank you in advance for your revisions.

Reviewer 4 Report

Overall an interesting and complex topic. It can be greatly improved with careful proofreading and rephrasing when need to make sure the concepts are expressed clearly. Please find more comments below:

Indicatevely:
"While Social media becomes very popular tool for sharing information " ->
"While Social media becomes A very popular tool for sharing information "

Rephrasing:
"By comparing the survey responses between US and Korean users in the social media, this study found out that ethical culture is positively, information privacy concerns are negatively associated with users’ information sharing behavior on social media while these size of impact of two facts are varied between two countries."

"In the real world, the definition of ‘culture’ is an ideational system, which means what people learn and how they behave, and the way people interpret their experiences and lead behavior."

Introduction:

The manuscript deals with a very complex concept, the one of culture.
In my opinion the introduction should be reorganised to provide a clear flow of thought. As is now, it passes from a general definition of culture, to mentions of culture and sound information, to end up in the definition of ethical culture which should come previously.

There are for sure many schools of thought but the definition of culture provided is a bit weak.
From a sociological perspective culture is a complex of languages, customs, beliefs, rules, arts, knowledge, and collective identities and memories developed by members of all social groups that make their social environments meaningful.

The definition of "ethical culture" should be provided by the first mention of the term.

Please explain further: "culture is one of the most important preconditions for sound information sharing in social media."

Not clear: "Most online ethical issues appear in the results of damage due to information and are linked to complex privacy issues."

"Eventually, people share their information on social media based 151 on mutual trust and mutual trust requires the belief that those that share information will not cause 152 damage to them as information is properly shared" People share their data also because of commodity and services, as proven by the well-known privacy paradox. GIVE CITATION

Not clear: "H1. Perceived ethical culture is positively associated with information sharing behaviors of social media users." This hypothesis assumes that ethical culture is increasing or descreasing with the sharing behaviour. I doubt this is what the author intends to express. I believe the author intended to say that information sharing is increased based on the presence of like-minded peers and perhaps homophilic attitudes.

Not clear:
"H3. The impact of ethical culture on information sharing behaviors on social media will be different with cultural difference."Different in which sense? "H4. The impact of information privacy concerns on information sharing behaviors on social media will be different with cultural difference" Different in which sense?

Related Works:

Important references missing:

- Triandis, H.C., 1994. Culture and social behavior.
- Fiske, J., 2010. Understanding popular culture. Routledge.
- Hofstede, G., 2001. Culture's consequences: Comparing values, behaviors, institutions and organizations across nations. Sage publications.
- Barnes, S.B., 2006. A privacy paradox: Social networking in the United States. First Monday.

"Their study argued that writing online movie reviews was affected by cultural differences such as differences in social norms and attitudes among 195 countries and indicated that significant results were identified through investigations with the US, China, and Singapore [23]."
The world values survey can be a useful resource to mention here.

Methodology

check 11 ref

It is mentioned that "All construct measurements are 217 modified in the context of social media and used with seven Likert scale."
Since this is a collection of items from different studies, the exact phrasing of all items asked should be reported here.
Please improve the caption of Table 1. Captions should be self-expanatory at all times.

Rather than the loadings I would expect to see basic statistics on the data would help the reader understand better the effects we are looking at. What is the distributions of each item? Perhaps an odds ratio on a logarithmic scale would be more adequate to show the effects.

"Eigen values" is one word.

What is the geographic representation of the US sample? If not representative this should be mentioned in the limitations section.

The methodology section should be greatly improved. Please explain thoroughly the methods used to derive the respective results. Also please thoroughly justify the choice of methods.

Results:

"Most respondents are college students who are frequently using the Internet as well as social media and males were 198 responses (50.9%) and females were 191 responses (49.1%)." what is the gender and age distribution per country?

"By conducting bootstrapping procedures," which procedures exactly?

I would expect a within and between country comparison right from hypothesis 1 perhaps based on anova. The same holds for the other hypothesis.

Discussion:

Interesting new study in line with your results:
- Kalimeri, K. and Tjostheim, I., 2020, July. Artificial Intelligence and Concerns About the Future: A Case Study in Norway. In International Conference on Human-Computer Interaction (pp. 273-284). Springer, Cham.

Round 2

Reviewer 1 Report

The discriminant validity is determined by three analyses.

First, the Fornell-Larcker criterion where the

square root of the AVE of each latent variable is higher than

its highest correlation with any other latent variable.

Second, the cross-loadings where an indicator’s outer

loading on a latent variable is higher than all its crossloadings

with other latent variables. Third, the heterotraitmonotrait

ratio of correlations (HTMT) where all values are

smaller than 0.9. These three analyses show the

existence of discriminant validity.

The authors confirmed that, they added he Fornell-Larcker criterion, but it is not added.

Reviewer 3 Report

Thank you for your improvements done. It looks like a strong submission, right now.

Reviewer 4 Report

I would like to thank the authors for their responses. I believe the manuscript has been significantly improved. 

Please proofread again the final manuscript since there are a few minor grammar mistakes.
